# Bacteriophage-related epigenetic natural and non-natural pyrimidine nucleotides and their influence on transcription with T7 RNA polymerase

Filip Gracias[1], Radek Pohl [1], Veronika Sýkorová[1] & Michal Hocek [1,2] ✉

DNA modifications on pyrimidine nucleobases play diverse roles in biology such as protection of bacteriophage DNA from enzymatic cleavage, however, their role in the regulation of transcription is underexplored. We have designed and synthesized a series of uracil 2'-deoxyribonucleosides and 5'-*O*-triphosphates (dNTPs) bearing diverse modifications at position 5 of nucleobase, including natural nucleotides occurring in bacteriophages, α-putrescinylthymine, α-glutaminylthymine, 5-dihydroxypentyluracil, and methylated or non-methylated 5-aminomethyluracil, and non-natural 5-sulfanylmethyl- and 5-cyanomethyluracil. The dNTPs bearing basic substituents were moderate to poor substrates for DNA polymerases, but still useful in primer extension synthesis of modified DNA. Together with previously reported epigenetic pyrimidine nucleotides, they were used for the synthesis of diverse DNA templates containing a T7 promoter modified in the sense, antisense or in both strands. A systematic study of the in vitro transcription with T7 RNA polymerase showed a moderate positive effect of most of the uracil modifications in the non-template strand and some either positive or negative influence of modifications in the template strand. The most interesting modification was the non-natural 5-cyanomethyluracil which showed significant positive effect in transcription.

Modified pyrimidine nucleobases play diverse roles in biology. 5-Methylcytosine (5mC)[1] is an important epigenetic modification in most eukaryotic genomes typically downregulating transcription. Its oxidized congeners[2,3], i.e., 5-hydroxymethylcytosine (5hmC)[4,5], 5-formylcytosine (5fC)[6] and 5-carboxycytosine (5caC)[7], that are formed through oxidation of 5mC by ten-eleven translocation (TET) enzymes[8,9] are intermediates in active demethylation[10–12] but also can play a role in regulation of transcription[13–15]. On the other hand, 5-hydroxymethyluracil (5hmU) is a very rare natural DNA modification occurring in human stem cells[16], cancer cells[17], or protozoan parasites[18], yet its biological role is not fully understood[19]. Most interestingly, in several bacteriophages 5hmU almost completely replaces thymine[20,21] and some bacteriophages even contain hypermodified pyrimidine bases[20,22], e.g., α-putrescinylthymine[23,24], α-glutaminylthymine[25], 5-dihydroxypentyluracil[26,27], 5-aminomethyl- and 5-aminoethyluracil[22], as well as glucosylated hydroxymethylcytosine[28,29]. Other interesting pyrimidine modifications have been observed in other organisms, e.g., base J in kinetoplastid protozoa[30–32] and 5-glycerylmethylcytosine in alga *Chlamydomonas*

*reinhardtii*[33,34]. Interestingly, 5-cyanomethyluridine was found as a modification in tRNA of *Haloferax Volcanii*[35]. There have been many studies on biosynthesis[36] of these hypermodified nucleotides and on their role in protection of virus DNA from cleavage by bacterial restriction endonucleases (REs)[37,38], but their role in regulation of transcription[39] is yet underexplored[20].

We have studied the influence of natural and non-natural nucleobase modifications on cleavage of DNA by REs and found[40,41] that some smaller modifications (i.e., hydroxymethyl or ethynyl) are often tolerated by RE that can still recognize and cleave the target DNA sequences and we have even shown proof-of-concept transient protection of DNA from RE cleavage[42,43]. We have also systematically studied the role of nucleobase modifications in regulation of transcription with bacterial RNA polymerase (RNAP) from *E. coli* and *B. subtilis* and found[44] that bulkier modifications typically completely inhibit transcription through blocking of the interaction of the RNAP and transcription factors with promoter sequences of the DNA. On the other hand, some small modifications, i.e., 7-deazapurines or 5-ethynyluracil are

[1]Institute of Organic Chemistry and Biochemistry, Czech Academy of Sciences, Flemingovo nam. 2, CZ-16000 Prague 6, Czech Republic. [2]Department of Organic Chemistry, Faculty of Science, Charles University, Hlavova 8, CZ-12843 Prague 2, Czech Republic. ✉e-mail: hocek@uochb.cas.cz

tolerated and allow transcription[44], whereas 5-hydroxymethyluracil[45], 5-hydroxymethylcytosine[45] or 5-ethyluracil[46] are not only tolerated but, surprisingly, they even significantly enhance transcription with *E. coli* RNAP. Also, some non-natural oxidized forms of 5-ethyl and 5-propylpyrimidines were tolerated and allowed transcription[46]. 5-Glucosyl-hmC was found to protect the DNA from RE cleavage but still gave transcription[47]. Photocaged 5hmU was used for switching transcription through photochemical release of 5hmU (triggering the transcription on) and phosphorylation by bacterial 5hmU DNA kinase (5HMUDK) that switched the transcription off again[48]. These findings suggest that the presence of 5hmU gives the bacteriophage DNA advantage in transcription, the enzymatic phosphorylation can be the bacterial response inactivating the viral DNA, while the glucosylation of 5hmC protects the viral DNA and keeps in transcriptionally active. The intriguing results with bacterial RNAP prompted us to study the influence of natural or non-natural pyrimidine modifications on in vitro transcription (IVT) with bacteriophage T7 RNA polymerase (T7 RNAP) which is used in enzymatic production of RNA through IVT[49], including RNA vaccines[50]. There has been two previous related studies[51,52] showing that small pyrimidine modifications (5-halouracil or 5-methyl- or 5-hydroxymethylcytosine) placed into the template (antisense) strand of the promoter do not prevent transcription. In this paper we present the synthesis of the modified nucleotides, enzymatic synthesis of DNA templates containing the modified nucleotides in either sense or antisense strand of the promoter or in both strands of the promoter or the whole sequence, as well as on the systematic study of the influence of the modifications on IVT with T7 RNAP (Fig. 1).

## Results

Similarly to our previous works[40–48], we intended to synthesize modified DNA templates through polymerase incorporation of 5-substituted pyrimidine 2′-deoxyribonucleoside triphosphates. We included some of the previously synthesized 5-hydroxymethyl-[45], 5-ethyl-, 5-hydroxyethyl and 5-acetylpyrimidine[46] nucleotides (Fig. 2), as well as several rare natural modifications, i.e., α-putrescinylthymine, α-glutaminylthymine, 5-dihydroxypentyluracil, and methylated or non-methylated aminomethyluracil, and related non-natural 5-sulfanylmethyl- and 5-cyanomethyluracil.

Firstly, we needed to synthesize the nucleoside intermediates where all highly nucleophilic groups needed suitable protection for the subsequent triphosphorylation. The synthesis of all started from thymidine, which upon protection and radical bromination provides the main intermediates—Ac-protected 5-bromomethyl-2′-deoxyuridine **1** or TBDPS-protected 5-bromomethyl-2′-deoxyuridine **2** (Fig. 3).

The acetyl-protected 5-bromomethyl-deoxyuridine intermediate (**1**) was used to prepare 5-(methylamino)methyl-2′-deoxyuridine (**dU**mm), 5-(dimethylamino)methyl-2′-deoxyuridine (**dU**dm) and 5-cyanomethyl-2′-deoxyuridine (**dU**cm) in analogy to previously published procedures[24,53–55]. Nucleophilic substitution reactions with appropriate nucleophiles, followed by immediate deprotection provided the target nucleosides in moderate yields (17% for **dU**mm, 20% for **dU**dm and 7% for **dU**cm, in 4 steps from thymidine). These relatively low yields are mainly due to the bromination reaction[55], which (in our hands) provided yields of only around 70% and the crude product could not be purified from the reaction mixture due its inherent high sensitivity to hydrolysis. Aminomethyl-2′-deoxyuridine (**dU**am) was prepared by the previously reported[54] catalytic hydrogenation of 5-azidomethyl-2′-deoxyuridine. Both **dU**am and **dU**mm needed to be protected by the reaction with ethyl-trifluoroacetate and Et3N in methanol. Similar protection was needed in case of known 5-sulfanylmethyl-2′-deoxyuridine[56] (**U**sm), which was prepared directly as an *S*-acetyl protected **dU**sm (**dU**asm) from TBDPS-protected bromo-derivative (**2**) and potassium thioacetate, followed by TBDPS deprotection by acidified TBAF (29% in 4 steps from thymidine).

Next, we sought to prepare three non-canonical nucleosides that were found in several bacteriophages—namely putrescine-, glutamic acid-, and diol-linked thymidines (**dU**put, **dU**glu and **dU**dhp, respectively)[24,25,27]. Initially, we intended to prepare **dU**glu and **dU**put by reductive amination of acetyl protected 5-formyl-2′-deoxyuridine with the respective amine or glutamic acid. However, a problematic imine formation combined with an undesired 1,4-addition of tested reductive agents on imine made us abandon this approach and instead we choose the same reaction pathway as mentioned above. Hence, the **dU**put was prepared in an analogy to the previously published procedure[24] but using Boc-protected putrescine as the nucleophile instead of reported monophthaloylputrescine (Fig. 4). Concomitant deprotection of both acetyl- and Boc- groups by acetyl chloride in dry methanol provided **dU**put (20% in 4 steps from thymidine) as the major product, which again needed to be TFA-protected before triphosphorylation. For **dU**glu, the nucleophilic substitution was performed using dimethylester of L-glutamic acid with Et3N in MeCN, instead of unprotected glutamic acid, which has shown insufficient reactivity and solubility in multiple reaction conditions. Attempted direct deprotection of both Ac- and Me- protecting groups on intermediate **5** resulted in cyclization of glutamic acid moiety to form 5-carboxy-γ-lactam modified nucleoside. Therefore, we performed a 3-step cascade that includes Boc-protection of the free amine, followed by prolonged complete deprotection of all acetyl and methyl groups by K2CO3 in H2O/MeOH mixture, followed by the final Boc-deprotection by TFA in DCM. This provided the desired **dU**glu in an

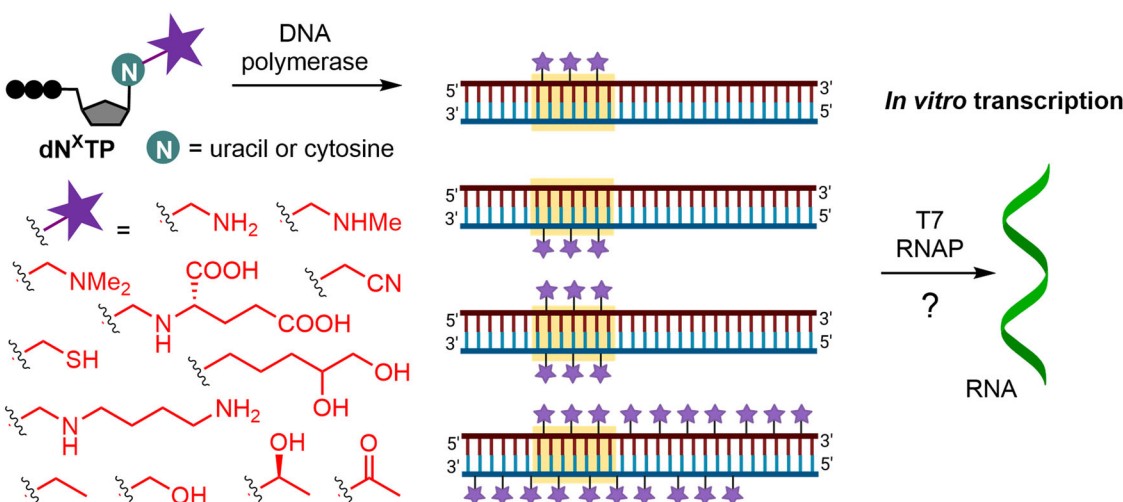

**Fig. 1 | Outline of the synthesis of modified DNA templates and study of the influence of pyrimidine modifications on IVT with T7 RNAP.** Partially created in BioRender. Gracias, F. (2024) BioRender.com/p33m837.

acceptable yield (9% in 6 steps from thymidine) (for NMR spectra of all prepared compounds see Supplementary Data 1, for IR spectra see Figs. S145–S155 in Supplementary Information).

To avoid problems with selectivity in phosphorylation of nucleoside **dU$^{dhp}$**, we have directly synthesized the corresponding nucleoside triphosphate (**dU$^{dhp}$TP**) using Sonogashira coupling conditions on 5-iodo-2′-deoxyuridine 5′-O-triphosphate with racemic 4,5-dihydroxypent-1-yne (Fig. 5), followed by Pd-catalyzed hydrogenation of the triple bond in analogy to previous work[57]. The desired **dU$^{dhp}$TP** was obtained and isolated in 46% yield (in 2 steps from dU$^I$TP).

All the above mentioned nucleosides were phosphorylated using slightly modified standard Ludwig conditions[58] with optional use of Proton Sponge, in some cases followed by immediate removal of protecting groups by aqueous ammonia (TFA removal in case of **dU$^{am}$TP**, **dU$^{mm}$TP** and **dU$^{put}$TP**; Ac removal in case of **dU$^{sm}$TP**) (Fig. 6). We also prepared two triphosphates with the protected aminomethyl and sulfanylmethyl groups (**dU$^{tfa}$TP** and **dU$^{asm}$TP**). The yields ranged from 3 to 30%. The lowest yields (3% for **dU$^{tfa}$TP** and 7% for **dU$^{am}$TP**) were achieved in triphosphorylation of **dU$^{tfa}$**, which showed numerous side-products, probably due to partial deprotection and subsequent reactivity of the primary amine. Another low yielding procedure (7%) was the triphosphorylation of **dU$^{glu}$TP** containing unprotected α-amino group of glutamic acid moiety, where the lower yield was compensated by shorter reaction sequence. All the desired **dU$^X$TPs** were prepared in sufficient quantities for the follow-up biochemical profiling and synthesis of modified DNA templates.

With the set of base-modified **dU$^X$TPs** in hand, we first tested them as substrates for DNA polymerases in primer extension (PEX) using a 5′-FAM-labelled 15-nt primer **Prim$^{15ON}$-FAM** and a 19-nt template **Temp$^{19ON\_T}$** encoding for incorporation of one modified deoxyuridine (for sequences, see Table S2 in Supplementary Information). All modified **dU$^X$** nucleotides were successfully incorporated into DNA in the PEX reactions using KOD XL DNA polymerase to produce 19-bp double-stranded DNA products **19DNA_U$^X$** that were visualized and characterized by both denaturing polyacrylamide gel electrophoresis (dPAGE) (Fig. 7A and Fig. S2 in Supplementary Information) and by LC-MS using UniDec deconvolution tool[59] (see Table S3 and Figs. S23–S46 in Supplementary Information). Only **dU$^{dm}$TP** showed sub-optimal substrate activity, but the full-length product was still observed. More demanding PEX reaction using a 31-bp template **Temp$^{31ON}$** encoding for four modifications (for sequences, see Table S4 in Supplementary Information) also worked for most modified **dU$^X$TPs** and the resulting modified **31DNA_U$^X$** were characterized by dPAGE (Fig. 7B and Fig. S3 in Supplementary Information) and LC-MS (Table S5 and Figs. S47–S70 in Supplementary Information). Only **dU$^{glu}$TP**, **dU$^{mm}$TP** were less efficient substrates but still gave full-length products, while **dU$^{put}$TP** and **dU$^{dm}$TP** unfortunately did not give any full-length

**Fig. 2 | Additional nucleoside triphosphates used in this study.** These nucleotides were previously prepared in our group.

**Fig. 3 | Synthetic overview of prepared nucleosides bearing small functional modification.** Reagents and conditions: *i*) Pd/C, H$_2$, EtOH, H$_2$O, 23 °C, 1.5 h; *ii*) TFAEt, Et$_3$N, MeOH, 23 °C, 16 h; *iii*) MeNH$_2$, Et$_3$N, MeCN, 0 °C, 15 min; *iv*) NH$_4$OH, MeOH, 23 °C, 2 h; *v*) Me$_2$NH, Et$_3$N, MeCN, −20 °C, 1.5 h; *vi*) KCN, DMF, 50 °C, 16 h; *vii*) AcSK, Et$_3$N, DMF, 75 °C, 1.5 h; *viii*) TBAF, AcOH, THF, 23 °C, 4 h.

**Fig. 4 | Synthetic overview of putrescine- and glutamic acid-modified thymidine.** Reagents and conditions: *i*) N-Boc-putrescine, Et₃N, DCM, 23 °C, 3 h; *ii*) AcCl, MeOH, 23 °C; *iii*) TFAEt, Et₃N, MeOH, 23 °C, 16 h; *iv*) dimethyl-L-glutamate HCl, Et₃N, DCM, 23 °C, 3 h; *v*) Boc₂O, MeOH, 23 °C, 48 h; *vi*) K₂CO₃, H₂O, MeOH, 23 °C, 24 h; *vii*) TFA, DCM, 23 °C, 35 min.

**Fig. 5 | Synthesis of dU^dhpTP.** Reagents and conditions: *i*) Pd(OAc)₂, CuI, TPPTS, Et₃N, 80 °C, 1 h; *ii*) H₂, H₂O, MeOH, 23 °C, 24 h.

products. The thiol-linked triphosphate **dU^smTP** gave a full-length product according to dPAGE but the product **31DNA_U^sm** could not be confirmed by MS—probably due to the oxidation of the thiol group. We tried to mitigate this by adding dithiothreitol (DTT) or tris(2-carboxyethyl)phosphine (TCEP) into the reaction and after purification also to the solution of purified DNA but it did not help. We also tried to use *S*-acetyl protected analog **dU^asmTP**, which could be incorporated into DNA and then deprotected by a solution of sodium hydroxide when needed. However, the *S*-acetyl group was quite unstable and prone to hydrolysis during standard PEX conditions.

Testing of the **dU^XTPs** in polymerase chain reaction (PCR) using a 98 bp template **98DNA** with 5′-FAM-labelled 20-nt primer **Prim^20ON-FAM** and 5′-Cy5-labelled 25-nt primer **Prim^25ON-Cy5** (for sequences, see Table S6 in Supplementary Information) showed that only few triphosphates can provide full length amplicons, namely **dU^dhpTP, dU^tfaTP, dU^cmTP, dU^asmTP** and **dU^smTP** (Fig. 7C and Fig. S4 in Supplementary Information). In some cases, this was possible only after optimizations, extra Mg²⁺ ions or extra DTT (see Section 2.4.3. in Supplementary Information for more information). It should be noted, that incorporation of **dU^cmTP** in PCR using a similar mixture of thermostable DNA polymerases was reported previously[60]. All the dNTPs with highly basic substituents (**dU^amTP, dU^mmTP, dU^dmTP** and **dU^putTP**) were poor substrates and did not give PCR amplicons. This can be explained by unfavorable repulsion of the protonated cationic substituent with positively charged basic arginine(s) and/or lysine(s) in the active site of the polymerase[61]. On the other hand, the TFA-protected **dU^tfaTP** was incorporated well even in PCR, where the corresponding free amine **dU^amTP** failed. LC-MS analysis of the PCR product (**98DNA_U^tfa**) showed partial deprotection (three to seven TFA groups cleavages detected, Fig. 7D; Table S7 and Figs. S137–S140 in Supplementary

Information). After a treatment with 50 mM NaOH for 1 hour, the follow-up LC-MS showed a mass corresponding to full length product of **dU^am**-modified DNA (**98DNA_U^am**) (Fig. 7D; Figs. S141–S144 in Supplementary Information). This shows a viable route to longer DNA modified with **U^am**, even though the unprotected **dU^amTP** is not an efficient substrate in more demanding polymerase reactions.

Next, we wanted to test all viable modified triphosphates as well as some previously prepared ones[45,46] (Fig. 2) in in vitro transcriptions with T7 RNA polymerase. We wanted to study the transcription efficiency (the amount of RNA produced by T7 DNA polymerase) when we introduce our modifications in the T7 promoter region of either sense strand, antisense strand, both strands and also if we modify the whole DNA template. To this end, we designed a 107 bp template with the T7 promoter region localized between 21ˢᵗ and 37ᵗʰ base pair. The templates were also modified with FAM on both 5′ ends and, at the 5′ end of the antisense strand by three 2′-OMe ribonucleotides, which help to prevent T7 RNA polymerase from non-templated additions[62]. This enabled us to prepare the modified templates using only enzymatic reactions and commercially available oligonucleotides. For each set of modified templates, a natural template was prepared in the same manner to be used as a reference.

To prepare templates with sense-modified promoter (**107DNA_S**), we first prepared modified 37-bp DNA (**37DNA**) containing T7 promoter region in a PEX reaction with KOD XL DNA polymerase using a 37-nt template **Temp^37ON-P** and 5′-FAM-labelled 20-nt primer **Prim^20ON-FAM** (for sequences, see Table S8 in Supplementary Information). All used modified **dN^XTPs** provided full-length PEX products, which were first confirmed by dPAGE (Fig. 8A and Fig. S5 in Supplementary Information) and LC-MS (Table S9 and Figs. S71–S102 in Supplementary Information)

**Fig. 6 | Synthesis of modified dNTPs through tri-phosphorylation of nucleosides.** Method A was used for the preparation of triphosphates without any deprotection step. Method B was used for the preparation of triphosphates with a protecting group, that needed to be removed post-triphosphorylation.

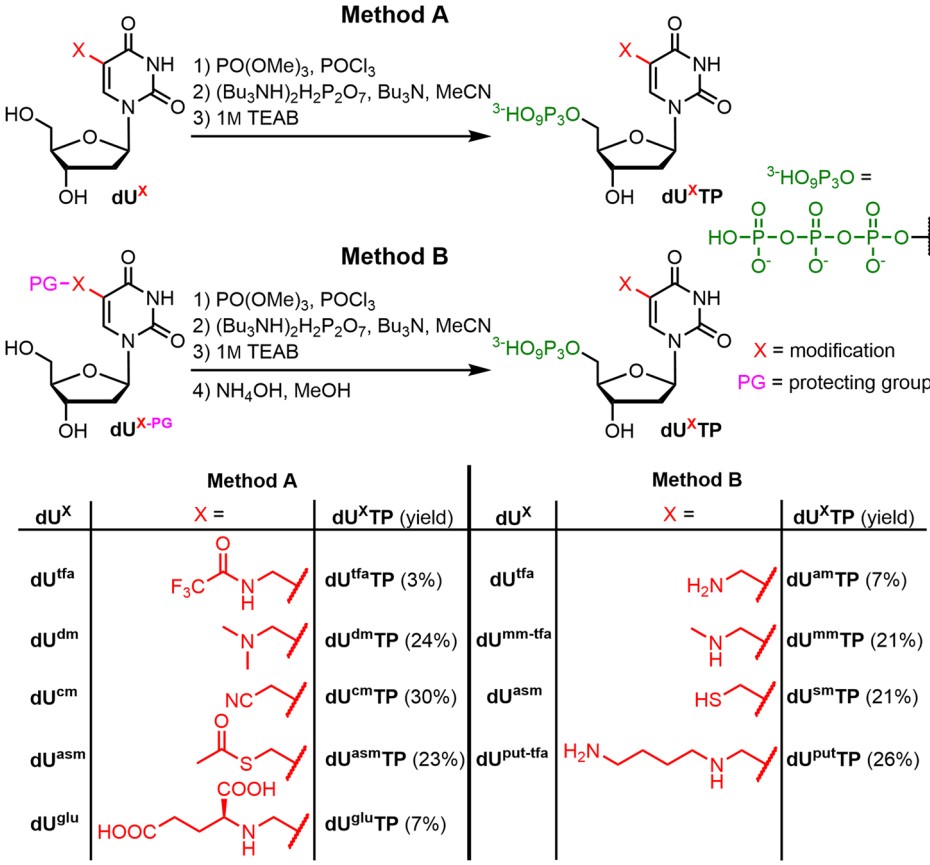

| dU^X | X = | dU^XTP (yield) | dU^X | X = | dU^XTP (yield) |
|---|---|---|---|---|---|
| **dU^tfa** | (trifluoroacetyl aminoethyl) | dU^tfaTP (3%) | **dU^tfa** | (aminoethyl) | dU^amTP (7%) |
| **dU^dm** | (dimethylaminoethyl) | dU^dmTP (24%) | **dU^mm-tfa** | (methylaminoethyl) | dU^mmTP (21%) |
| **dU^cm** | (isocyano ethyl) | dU^cmTP (30%) | **dU^asm** | (thiol ethyl) | dU^smTP (21%) |
| **dU^asm** | (acetylthio ethyl) | dU^asmTP (23%) | **dU^put-tfa** | (aminobutyl aminoethyl) | dU^putTP (26%) |
| **dU^glu** | (glutamic acid amide) | dU^gluTP (7%) | | | |

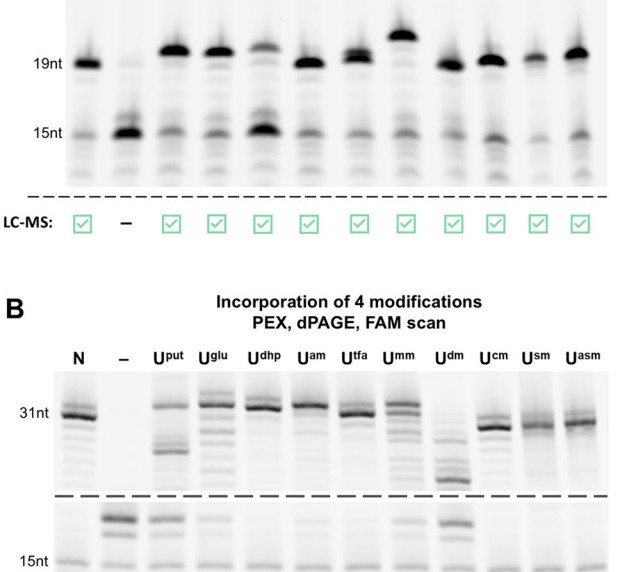

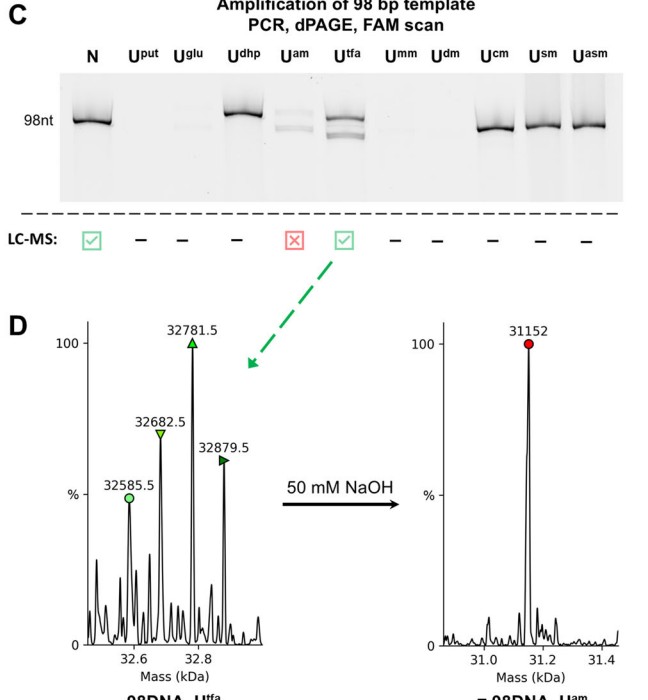

**Fig. 7 | Summary of nucleoside triphosphate testing in enzymatic incorporations. A** Incorporation of 1 modified **dU^XTP** by PEX to produce **19DNA_U^X**; **B** Incorporation of 4 modified **dU^XTP** by PEX to produce **31DNA_U^X**; **C** Amplification of **98DNA** by PCR with modified **dU^XTP**; **D** LC-MS analysis of **98DNA_U^tfa** and its product after a treatment with aqueous NaOH. All enzymatic reactions use KOD XL polymerase. **N** = natural DNA, **U^X** refer to modified nucleobases. LC-MS data deconvoluted using UniDec.

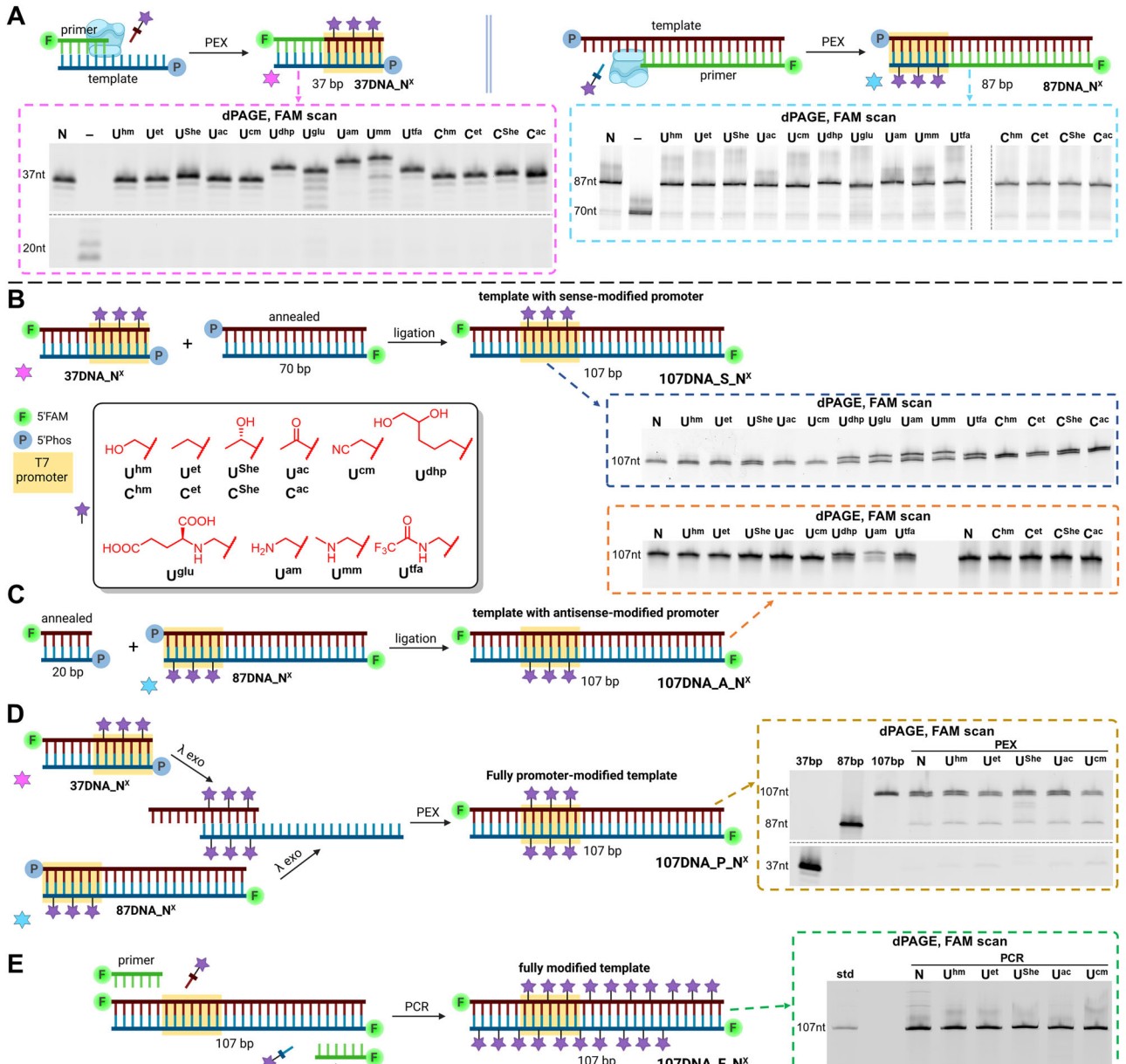

**Fig. 8 | Preparation of modified DNA building blocks and templates for transcription testing. A** Preparation of modified DNA building blocks **37DNA_N^X** (pink color) and **87DNA_N^X** (cyan color) by PEX reactions, using KOD XL polymerase; **B** preparation of templates with sense-modified promoter via ligation by T4 ligase; **C** preparation of templates with antisense-modified promoter via ligation by T4 ligase; **D** preparation of templates with fully modified promoter via partial λ-exo digestion of products in (A), followed by PEX reaction, using Vent (exo-) polymerase; **E** preparation of fully modified templates by PCR, using KOD XL polymerase. **N** = natural DNA, **U^X** refer to modified nucleobases. Partially created in BioRender. Gracias, F. (2024) BioRender.com/l73g378.

and then ligated to a 70-bp dsDNA by T4 DNA ligase, using optimized conditions (Fig. 8B) (for more information, see Section 2.7.1. in Supplementary Information). The desired products were then purified by gel extraction from an agarose gel. Previously in our lab we noticed that after preparative ligation reaction, a small amount of partially ligated (nicked) product can be generated. This nicked product can exist in two variants (depending on which strand is not ligated) and is purified from agarose gel together with the desired fully ligated template. Therefore, we performed a dPAGE analysis to check for presence of these nicked products. The analysis has shown presence of minimal amount of nicked products (see Fig. S9 in Supplementary Information for full gel) and so we proceeded forward by quantifying the amount of produced **107DNA_S** templates based on 6-FAM signal (for more information, see Section 2.8.1. in Supplementary Information).

Templates with antisense-modified promoter (**107DNA_A**) were prepared in a similar manner by first preparing modified 87-bp (**87DNA**) by PEX using an 87-nt template **Temp^{87ON}-P** and either 5′-FAM-labelled 70-nt primer **Prim^{70ON}-FAM** (in case of U^X) or 68-nt primer **Prim^{68ON}-FAM** (in case of C^X) (for sequences, see Tables S10 and S11 in Supplementary Information). The full-length products were confirmed by dPAGE (Fig. 8A and Figs. S6 and S7 in Supplementary Information) and (most of them also) by LC-MS (Table S12 and Figs. S103–S132 in Supplementary Information) and then ligated to 20-nt dsDNA (Fig. 8C). The dPAGE analysis after ligation showed some nicked products but still in minor amounts (see Fig. S10 in Supplementary Information for full gel). In this case, **107DNA_A_U^{mm}** and **107DNA_A_U^{glu}** could not be prepared due to insufficient incorporation of the corresponding modified nucleotides in PEX reaction and were therefore not tested.

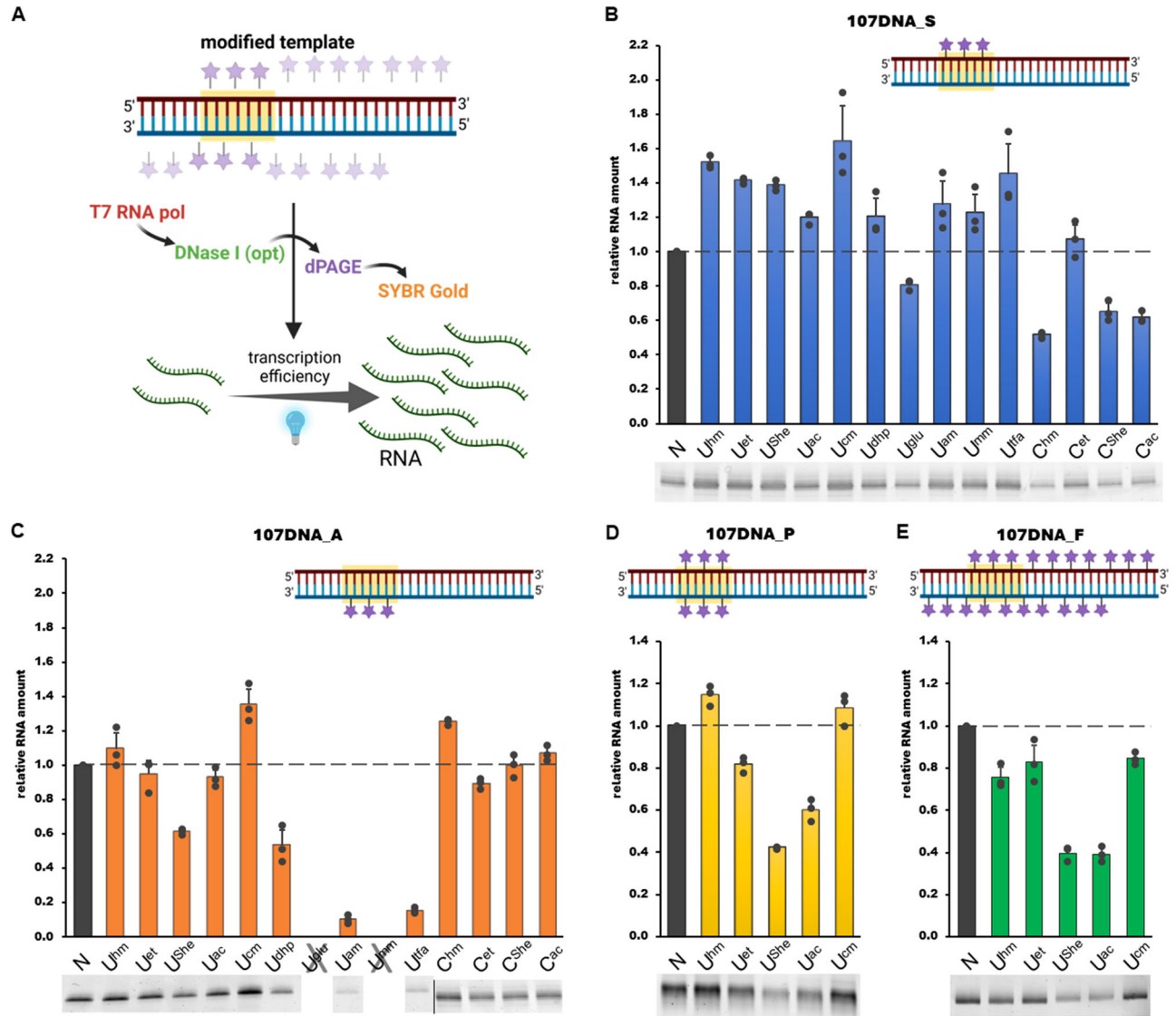

**Fig. 9 | Scheme of transcriptional testing and the follow-up results. A** General scheme for transcriptional testing; **B** results from templates with sense-modified promoter; **C** results from templates with antisense-modified promoter; **D** results from templates with fully-modified promoter; **E** results from fully modified templates. Natural DNA (N) was set to 100%. *n* = 3, error bars represent ±SD. Partially created in BioRender. Gracias, F. (2024) BioRender.com/q45b994; BioRender.com/p33m837.

We also prepared templates with modified promoter in both strands (**107DNA_P**) and fully modified templates containing the modifications in both strands in the whole sequence (**107DNA_F**). Here we used only a subset of modified nucleobases, namely **U^hm**, **U^et**, **U^She**, **U^ac** and **U^cm** (Fig. 8D, E), whose **dN^XTPs** were good substrates in PEX and PCR and which gave some positive effect in transcription in our preliminary experiments. The promoter-modified templates (**107DNA_P**) were prepared by a Lambda exonuclease digestion of 5'-phosphorylated strand in **87DNA** and **37DNA** to produce modified single stranded DNA (ssDNA; **87ON** and **37ON**) (Fig. S8 in Supplementary Information). These two ssDNA were then combined and elongated in a double PEX reaction by Vent(exo⁻) DNA polymerase (Fig. 8D) (see Fig. S11 in Supplementary Information for full gel). Fully modified templates **107DNA_F** were prepared by standard PCR with KOD XL DNA polymerase using the natural **107DNA_S** used as a template with 5′-FAM-labelled primers **Prim^20ON-FAM** and **Prim^16ON-FAM** (for sequences, see Table S17 in Supplementary Information), utilizing slightly modified conditions (Fig. 8E) (see Fig. S12 in Supplementary Information for full gel).

With these modified templates in hand, we moved to IVT testing. We chose to use T7 transcription conditions used previously in our lab[63], with 10 ng of our templates for 1 hour, followed by DNA template digestion by DNase I for additional 15 min. In order to avoid working with radioactive isotopes (γ-³²P GTP labelling of the RNA), we opted to visualize the synthesized RNA after gel electrophoresis by SYBR Gold staining (Fig. 9A). The fluorescent signal of RNA produced by natural template (N) was used as a control and assigned a value of 100%, to which signals of RNA produced from modified DNA templates were compared (Fig. 9). Each template was used for three separate transcription reactions and the results were averaged.

Firstly, we tested IVT with templates containing modifications in sense-strand of promoter region (**107DNA_S**) and found that most of the tested modifications increased the amount of produced RNA by 20–65%, with **U^cm** being the modification with the most stimulating effect (Fig. 9B) (see Figs. S13 and S14 in Supplementary Information for full gel and graph). Only in case of **U^glu** and cytidine-modified **C^hm**, **C^She** and **C^ac** we saw some decrease of RNA synthesis (81%, 52%, 65% and 62%, respectively). Testing of templates with antisense-modified promoter **107DNA_A** showed that

$U^{cm}$ was again the most transcription stimulating modification (135% compared to natural template), followed by $C^{hm}$ (125%) (Fig. 9C) (see Figs. S15–S17 in Supplementary Information for full gels and graph). The other modifications showed either negligible effect (90–110% compared to natural template) or stronger suppression effect (61% for $U^{She}$, 53% for $U^{dhp}$, 10% for $U^{am}$ and 16% for $U^{tfa}$). Modification $U^{tfa}$ was particularly interesting, since it provided a stimulatory effect in sense-strand, but very strong inhibition in antisense-strand.

IVT reactions with templates modified in both strands (**107DNA_P** and **107DNA_F**) were performed without subsequent DNase I treatment and instead the reaction was stopped by EDTA, followed by immediate denaturation to avoid an undesirable blocking of template digestion by one or more modifications[64]. Templates with fully modified promoter region (**107DNA_P**) showed in all cases similar or slightly lowered level of RNA production, compared to partially modified promoter templates (Fig. 9D) (see Figs. S18 and S19 in Supplementary Information for full gel and graph). Only $U^{hm}$ and $U^{cm}$ modifications showed a minor stimulatory effect on IVT (114% and 109%, respectively). The fully modified templates **107DNA_F** in all cases gave lower transcription compared to non-modified template (Fig. 9E) (see Figs. S20 and S21 in Supplementary Information for full gel and graph) but the results were not too different from the promoter-modified templates suggesting minimal effect of these modifications outside of the promoter region.

Finally, we wanted to assess the fidelity of IVT with fully modified templates **107DNA_F** through sequencing of the produced RNA (**70RNA_F**). Therefore, we prepared samples for the next generation sequencing (NGS) by reverse transcription of purified **70RNA_F** from IVT, using M-MLV reverse transcriptase with target-specific primer **Prim_RT$^{18ON}$** and template switching oligo **TSO$^{30ON}$**, followed by PCR-based amplicon library preparations (see Section 2.9. in Supplementary Information for full description). Indexed libraries were pooled and sequenced on Illumina NovaSeq with an output of 2 millions of 2 ×150 bp paired-end reads per sample. The results (see Figs. S156–S162 in Supplementary Information) show high transcription fidelity with templates **107DNA_F** containing natural T, or modified $U^{hm}$, $U^{et}$, $U^{She}$ or $U^{cm}$ nucleobases. Only in case of template containing the $U^{ac}$ modification, the resulting RNA contained several partial (4–5%) A → G mutations (mainly positions 13, 16, 25, 43). Apparently, these mutations are caused by mis-incorporations due to mutagenic minor base-pairing between $U^{ac}$ and G, similarly as has been previously reported for 5-formyluracil[65]. However, we cannot rule out the possibility that part of these mutations could have been introduced into the DNA during PCR with modified **dU$^{ac}$TP**.

## Discussion

We have synthesized a series of **dU$^{X}$TP**s derived from bacteriophage nucleotides and several related non-natural 5-substituted nucleosides and **dU$^{X}$TP**s. The **dU$^{X}$TP**s bearing strongly basic amino groups or glutamic acid, as well as oxidizable thiol were rather poor substrates for DNA polymerases, but in most cases they could be at least used in PEX synthesis of modified ssDNA. In PCR, most of them do not give good amplification. This is probably because of unfavorable interactions of the positively charged substituents with cationic amino acids in the active site of the enzyme. The comparison of our results with biosynthesis of viral DNA containing modified nucleobases shows a distinct correlation. The modifications that were easily incorporated even in PCR using modified **dU$^{X}$TP**s ($U^{hm}$, $U^{dhp}$) are biosynthetically available directly as nucleoside triphosphates, whereas $U^{glu}$ and $U^{put}$, which failed to give full-length amplicon in PCR from their corresponding **dU$^{X}$TP**s, are introduced into DNA via post-synthetic transformations of $U^{hm}$[20].

We have successfully designed and executed the synthesis of DNA templates containing the modified pyrimidine nucleotides (including several previously reported[45,46] modified 5-substituted uracil and cytosine nucleotides) in either non-template (sense), template (antisense) or in both strands of the promoter, as well as in the whole sequence and systematically studied the transcription with T7 RNAP. The effect of pyrimidine

modifications on IVT with T7 RNA polymerase showed some similarities and some differences compared to previously reported works[45,46,66] on the effect on transcription with bacterial RNAP. Similarly to the bacterial RNAP, some pyrimidine modifications showed a stimulatory effect on IVT with T7 RNAP, but this effect was significantly weaker (max. ca 170%) compared to *E. coli* RNAP, where some of the modifications showed more than two-fold increase in transcription (350% for $U^{hm}$, 250% for $C^{hm}$ and 200% for $U^{et}$)[45,46]. The strongest positive effect of most of the uracil modifications and a weak negative effect of cytosine modifications was observed in the non-template (sense) strand of DNA template. On the other hand, the modifications in the coding (antisense) strand showed higher differences where some nucleobases bearing small neutral modifications had slightly stimulating ($U^{cm}$, $C^{hm}$, $U^{hm}$) or neutral ($U^{et}$, $C^{et}$, $U^{ac}$, $C^{ac}$) effect, whereas bulkier ($U^{dhp}$, $U^{tfa}$) or charged ($U^{am}$) modifications showed moderate to strong inhibiting effects. Several previous studies[52,67–71] and X-ray structure[72] of T7 RNAP showed that specific major-groove interactions are more important in the template strand which is in accord with our findings that some bulkier or charged modifications may disrupt these interactions and inhibit transcription. Conversely, the non-template (sense) strand apparently offers more freedom for accommodation of modifications which are generally better tolerated and some of them may even positively contribute to the interactions with the polymerase and thus enhance the transcription. However, a deeper mechanistic and structural biology study would be needed to fully understand the effects.

Unlike in the bacterial IVT[45,46], the modifications in both strands has only minor effect on T7 RNAP IVT and there was no increase in IVT in fully modified templates. The bacteriophage-derived natural modified pyrimidine nucleobases showed weak positive ($U^{am}$, $U^{mm}$ and $U^{dhp}$) or negative ($U^{glu}$) effect in the sense-strand, but strong negative effect in the antisense strand ($U^{am}$ and $U^{dhp}$). Surprisingly, the non-natural 5-cyanomethyluracil ($U^{cm}$) showed the strongest stimulatory effects in most cases and we can envisage some biotechnological applications of this nucleotide in more efficient IVT synthesis of RNA with T7 RNAP. The influence of these modifications on bacterial and eukaryotic transcription that could shed more light into their biological role will be subject of a separate study.

## Methods

Complete experimental part including synthesis of all nucleosides, nucleoside triphosphates and their characterization, and also of all enzymatic procedures, their analysis, DNA template preparations, in vitro transcriptions, and others are given in the Supplementary information.

### Example of triphosphorylation reaction–preparation of 5-cyanomethyl-2'-deoxyuridine-5'-O-triphosphate, triethylammonium salt (dU$^{cm}$TP)

**dU$^{cm}$** (40 mg, 0.150 mmol) was dried on high vacuum for 16 hours and then suspended in PO(OMe)$_3$ (0.5 mL), stirred at 23 °C for 10 min and then cooled to 0 °C. POCl$_3$ (16 μL, 0.18 mmol) was added dropwise over 1 min and the reaction was stirred at 0 °C for 4 hours. Then, Bu$_3$N (178 μL, 0.75 mmol) and chilled 0.5 M solution of (Bu$_3$N)$_2$H$_2$P$_2$O$_7$ in MeCN (1.2 mL, 0.6 mmol) were added and the reaction was stirred at 0 °C for 30 min and then 15 min at 23 °C. The reaction was finished by adding 1 M TEAB (2 mL) and stirred for 15 min. The mixture was diluted with H$_2$O, evaporated, diluted again with H$_2$O (5 mL) and lyophilized for 16 hours. The lyophilizate was dissolved in H$_2$O (22 mL) and separated on ion-exchange HPLC (0 to 100% of 800 mM TEAB in H$_2$O, Sepharose DEAE Fast Flow). The appropriate fraction were collected, evaporated, dissolved in buffer A and injected into HPLC (0 to 30% of buffer B in buffer A, Kinetex EVO C18). The pure triphosphate **dU$^{cm}$TP** (36.9 mg, 30%) was obtained as a hydroscopic amorphous solid.

### Preparation of 37DNA, 37DNA_U$^{X}$ and 37DNA_C$^{X}$ by primer extension reaction

The reaction mixture (100 μL) contained primer **Prim$^{20ON}$-FAM** (2.5 μM), template **Temp$^{37ON}$-P** (2.5 μM), natural dNTPs (dATP, dGTP and either

dCTP or dTTP, 200 µM each), appropriate **dU$^X$TP** or **dC$^X$TP** (for natural DNA either dTTP or dCTP, 200 µM; 300 µM for **dU$^{ac}$TP**; 400 µM for **dU$^{glu}$TP, dU$^{am}$TP, dU$^{mm}$TP**), KOD XL polymerase reaction buffer (10 µL) and KOD XL DNA polymerase (5 U; 7.5 U for **dU$^{glu}$TP, dU$^{am}$TP, dU$^{mm}$TP**). The reactions were then subjected to the following thermal cycler program: 95 °C for 5 min, followed by 55 °C for 10 min and then 72 °C for 5 min. The reactions were purified using QIAquick nucleotide removal kit (eluted in 30 µL of 5 mM Tris-Cl buffer, pH = 8.5).

## DNA analysis by LC-MS

LC-MS measurements of DNA were measured on Agilent 1920 Infinity II BIO system with MSD XT mass spectrometer equipped with ESI ion source, using Phenomenex Biozen 1.7µm Oligo 50 ×2.1 mm column together with buffer A (300 mM HFIP + 15 mM TEA in $H_2O$) and buffer B (300 mM HFIP + 15 mM TEA in MeOH). Gradients and flowrates were variable. Aquired spectra were deconvoluted using UniDec program with a deconvolution resolution of ± 0.5 Da.

## Ligation reaction procedure

The reaction mixture (50 µL) contained two appropriately 5'-phosphorylated dsDNAs (0.8 µM each), T4 DNA ligase (3200 U) and 2X Quick ligation buffer (132 mM Tris, pH = 7.6, 20 mM $MgCl_2$, 2 mM DTT, 2 mM rATP, 15% PEG 6000; 25 µL). The reaction was incubated at 23 °C for 30 min. After the reaction, samples were first pre-purified by QIAquick PCR purification kit (eluted in 30 µL of 5 mM Tris-Cl buffer, pH = 8.5). 6X loading dye (10 µL) was then added to the pre-purified samples and each sample was loaded onto 3% agarose gel (SERVA agarose for PCR, molecular biology grade) and run at 5 V/cm for 2 h. DNA was visualized by blue light transilluminator and the DNA of the desired length was cut out of the gel and purified on Qiagen MinElute columns, but using buffers and protocol from E.Z.N.A. Gel Extraction Kit (eluted in 15 µL of 5 mM Tris-Cl buffer, pH = 8.5). The products were then analyzed on denaturing PAGE to confirm the length of the products as well as the presence of only partially ligated DNA. This partially ligated DNA is visible as a residual FAM signal of the used dsDNA for the ligation.

## Transcription reaction using 107DNA_S and 107DNA_S_U$^X$/ 107DNA_S_C$^X$ containing sense-modified promoter

In vitro transcription reaction (20 µL) contained T7 transcription buffer (4 µL), $MgCl_2$ (25 mM), Triton X-100 (0.1%), rNTPs mix (2 mM), RiboLock RNase Inhibitor (20 U), either natural or modified DNA template (**107DNA_S, 107DNA_S_U$^X$** or **107DNA_S_C$^X$**; 10 ng, prepared according to Section 2.7.2.) and T7 RNA polymerase (20 U). Once the template was added, the reactions were incubated at 37 °C for 1 h. DNase I (2 U) was then added and the reactions were further incubated at 37 °C for 15 min, followed by the addition of stop solution (40 µL) and $H_2O$ (20 µL). The samples were denatured (65 °C for 10 min), chilled on ice and 5 µL was loaded onto 12.5% denaturing PAGE which was run at 25 mA for 50 min. The gel was stained with SYBR Gold (diluted to 1X in 1X TBE buffer) for 15 min, scanned (Fig. S13 in Supplementary Information) and the relative amount of RNA quantified. The whole procedure was done in triplicate and the results were averaged.

## Data availability

Detailed procedures and data are given in Supplementary Information. NMR data are given in Supplementary Data 1. Primary data are available from repository https://doi.org/10.48700/datst.r2a7v-fw652.

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

## Acknowledgements
The work was funded by the Ministry of Education, Youth and Sports of the Czech Republic OPJAK grant RNA for therapy (CZ.02.01.01/00/22_008/0004575) co-financed by the EU. Ph.D. scholarship for F.G. from the Department of Organic Chemistry of the University of Chemistry and Technology Prague is also acknowledged. The authors thank NGS facility of IOCB Prague for the help with NGS RNA sequencing.

## Author contributions
F.G. and M. H. designed the study, analyzed results and wrote the paper. F.G. performed the chemical synthesis and biochemical study. R.P. measured and characterized NMR spectra. V. S. and M. H. supervised the study and secured funding.

## Competing interests
The authors declare no competing interests. Michal Hocek is a Guest Editor for Communications Chemistry's Nucleic Acid Chemistry Collection, but was not involved in the editorial review of, or the decision to publish this article.
