## [Peer review file · Communications Chemistry]

Bacteriophage-related epigenetic natural and non-natural pyrimidine nucleotides and their influence on transcription with T7 RNA polymerase

Corresponding Author: Professor Michal Hocek

Version 0:

Reviewer comments:

Reviewer #1

(Remarks to the Author)

The manuscript describes the synthesis of a collection of natural rare nucleotides modified at the 5-position of pyrimidines (primarily dU and some dC) found in bacteriophages, their enzymatic incorporation into dsDNA, and a quantitative study of their influence on in vitro transcription with T7 RNA polymerase. While it is well known that these hypermodified nucleotides protect viral DNA from cleavage by bacterial restriction endonucleases, their role in transcriptional regulation remains underexplored. The tools developed in this study could help further elucidate this potential biological function.

The authors propose synthesizing modified DNA duplexes through polymerase incorporation of 5-substituted pyrimidine 2'-deoxyribonucleoside triphosphates and primer extension reaction. They report the synthesis of 18 natural and non-natural bacteriophage nucleotides, 8 of which had been described in previous studies, while 10 were synthesized according to known procedures using 5-iodo-dU or 5-bromomethyl-dU. The modified nucleoside triphosphates building blocks were prepared following Ludwig conditions. The synthetic procedures are well-described, and the products are well-characterized.

The incorporation of these modified nucleotides into DNA duplexes by a DNA polymerase is evaluated through increasingly complex models, including a PCR amplicon of around 100 base pairs obtained via a ligation step. Duplexes were obtained containing a T7 promoter region with modified pyrimidines on either the sense strand, the antisense strand, or both to assess the impact of these biochemical modifications on transcription efficiency. The resulting products are carefully characterized, demonstrating that the described method can be used to obtain synthetic DNA duplexes with the hypermodified nucleosides of interest.

Finally, the authors demonstrate the impact of these modifications on T7 polymerase activity by quantifying RNA production using a fluorescent assay. The results show either positive or negative effects depending on the strand containing the hypermodified bases.

This study is well conducted and interesting, but lacks a more in-depth discussion of the results. The manuscript leaves us somewhat frustrated and a deeper discussion of the biological implications based on existing knowledge is expected. For example, what is already known about the impact of modifications on the sense or antisense strand for other known epigenetic modifications on transcription activity? Could we draw any rational conclusions regarding T7 polymerase recognition based on its binding site? Although a full biological study is not required here, a more detailed discussion proposing explanations for these observations would greatly enrich the manuscript and help readers better understand the broader context and significance of the results.

Regarding the incorporation of synthetic nucleotide triphosphates by DNA polymerase, a discussion about the biosynthesis pathways of thymidine hypermodifications should be added to clarify the obtained results. Some DNA modifications naturally occur through different mechanisms, either before or after DNA polymerization. In this study, all modifications are introduced as triphosphate building blocks. It would be interesting to specify at which stage the pyrimidine modification takes place in cell among those studied here. If some of these modifications naturally occur post-polymerization, could that explain why they are less efficient substrates for DNA polymerase? Furthermore, do we know if such modified duplexes have already been synthesized from duplexes containing 5hmdU using specific enzymes involved in such hypermodification? Emphasizing this synthetic aspect would also further enhance the impact and significance of the study.

Minor: a reference is lacking in Figure 1 (dT_{N3} synthesis)

Reviewer #2

(Remarks to the Author)

In this manuscript, the authors synthesized a series of dUXTPs derived from unnatural 5-substituted nucleosides, and performed polymerase assays (PEX, PCR) for these synthetic substrates using Vent and KOD DNA polymerases. Furthermore, they performed transcription assays using T7 RNAP with polynucleotides containing 5-substituted nucleosides as templates.

Although the synthesis of these substrates is generally not easy, the substrates they synthesized this time have been satisfactorily identified by NMR and MS.

Accordingly, the reviewer recommends that the manuscript would be published with the following revisions:

-Assays of 5-substituted pyrimidine nucleoside triphosphates using thermostable DNA polymerases have already been reported (NAR, 2006, 5383), so comparisons and discussion are warranted in this paper (5-cyanomethyluridine/cytidine derivatives are also described in that report). Furthermore, the paper should also be cited.

-Regarding 5-cyanomethylpyrimidine derivatives in microorganisms, a previous report (RNA. 2014, 177) should be cited and discussed.

-The sequences of poly-ribonucleotides generated by T7 RNAP should be statistically analyzed and the fidelity (accuracy) should be discussed (e.g., see JACS, 2020, 21530).

-Polymerases used in the experiments should be described in the footnotes of the figures.

Reviewer #3

(Remarks to the Author)

The manuscript by Gracias et al. describes a continuation of previous work by the group on enzymatic synthesis of DNA templates containing novel and biologically relevant modified nucleotides. Subsequently the authors evaluated the influence of such modifications on the IVT with T7 RNAP. The work is highly important to the scientific community, with an emphasis on nucleic acid (bio)chemists and chemical biologists. The experiments are carried out well. I recommend for publication pending the revisions outlined below

1. Small molecule characterization should include 5 characterization techniques. The authors included ¹H, ¹³C and HRMS which are critical, but two additional techniques such as IR, UV, elemental analysis should be included for all novel compounds

2. Scheme 1: missing Pd/C for i

3. Page 5 line 102, I could not find dU_{sm}, is this dU_{asm}? If not, a chemical structure should be drawn in one of the figures or schemes for reference (note that Scheme 4 might be too far in the text for proper reference)

4. I am interested in the concomitant deprotection of Boc and Ac in MeOH. It would seem to me that upon addition of AcCl, the amine beta to the nucleobase would undergo acylation. The amine is more nucleophilic than methanol, and I would expect major by product formation with this step. Can the authors comment on this?

5. Page 7: authors mention low yield for certain triphosphates. Is there a reason for this? Can the authors offer a bit more information as to why this could be the case for these compounds.

6. An additional Figure should be generated for the main text concerning the initial primer extension experiments (including the generation of the 107 bp template). It is difficult to follow the discussion of about 1.5 pages, when always referring to the supporting information.

7. Can the authors provide more explanation concerning the U_{tfa} providing enhanced sense-strand activity and strong inhibition on antisense-strands?

8. There seems to be inconsistency in the abbreviations used for the chemical modifications. E.g. what is the difference between U_{cm} and dU_{cm}. One would logically think that the one with the "d" is DNA and the U alone is RNA. Is this the case? Since there are so many abbreviations in the text, the authors should really try to clarify this terminology.

Version 1:

Reviewer comments:

Reviewer #1

(Remarks to the Author)

The authors have made appropriate revisions to the manuscript and are congratulated on the advances reported.

Reviewer #2

(Remarks to the Author)

The manuscript has properly been revised.

Reviewer #3

(Remarks to the Author)

The authors have addressed all my comments (and other reviewer comments, in my opinion).

Response to reviewers and list of changes

COMMSCHEM-24-0389 Bacteriophage-related epigenetic natural and non-natural pyrimidine nucleotides and their influence on transcription with T7 RNA polymerase

Reviewers' comments:

Reviewer #1 (Remarks to the Author):

manuscript describes the synthesis of a collection of natural rare nucleotides modified at the 5-position of pyrimidines (primarily dU and some dC) found in bacteriophages, their enzymatic incorporation into dsDNA, and a quantitative study of their influence on in vitro transcription with T7 RNA polymerase. While it is well known that these hypermodified nucleotides protect viral DNA from cleavage by bacterial restriction endonucleases, their role in transcriptional regulation remains underexplored. The tools developed in this study could help further elucidate this potential biological function.

The authors propose synthesizing modified DNA duplexes through polymerase incorporation of 5-substituted pyrimidine 2'-deoxyribonucleoside triphosphates and primer extension reaction. They report the synthesis of 18 natural and non-natural bacteriophage nucleotides, 8 of which had been described in previous studies, while 10 were synthesized according to known procedures using 5-iodo-dU or 5-bromomethyl-dU. The modified nucleoside triphosphates building blocks were prepared following Ludwig conditions. The synthetic procedures are well-described, and the products are well-characterized.

The incorporation of these modified nucleotides into DNA duplexes by a DNA polymerase is evaluated through increasingly complex models, including a PCR amplicon of around 100 base pairs obtained via a ligation step. Duplexes were obtained containing a T7 promoter region with modified pyrimidines on either the sense strand, the antisense strand, or both to assess the impact of these biochemical modifications on transcription efficiency. The resulting products are carefully characterized, demonstrating that the described method can be used to obtain synthetic DNA duplexes with the hypermodified nucleosides of interest.

Finally, the authors demonstrate the impact of these modifications on T7 polymerase activity by quantifying RNA production using a fluorescent assay. The results show either positive or negative effects depending on the strand containing the hypermodified bases.

This study is well conducted and interesting, but lacks a more in-depth discussion of the results. The manuscript leaves us somewhat frustrated and a deeper discussion of the biological implications based on existing knowledge is expected. For example, what is already known about the impact of modifications on the sense or antisense strand for other known epigenetic modifications on transcription activity? Could we draw any rational conclusions regarding T7 polymerase recognition based on its binding site? Although a full biological study is not required here, a more detailed discussion proposing explanations for these observations would greatly enrich the manuscript and help readers better understand the broader context and significance of

the results.

Response: we thank the reviewer for this valuable suggestion – there were only two older papers reporting the tolerance of T7 RNAP to some modifications in the coding strand of the promoter – they had very different set up and hence a direct comparison is not relevant. However, we added a sentence and citations to the Introduction. On the other hand, there were several studies showing that the interactions of T7 RNAP with the template strand of the promoter are more important than with the non-template strand – which is nicely in accord with our results. So we extended the discussion accordingly.

Revision:

We added the following sentence to Introduction: There has been two previous related studies^{51,52} showing that small pyrimidine modifications (5-halouracil or 5-methyl- or 5-hydroxymethylcytosine) placed into the template (antisense) strand of the promoter do not prevent transcription.

We added the following part to the Conclusion: On the other hand, the modifications in the coding (antisense) strand showed higher differences where some nucleobases bearing small neutral modifications had slightly stimulating (\mathbf{U}^{cm} , \mathbf{C}^{hm} , \mathbf{U}^{hm}) or neutral (\mathbf{U}^{et} , \mathbf{C}^{et} , \mathbf{U}^{ac} , \mathbf{C}^{ac}) effect, whereas bulkier (\mathbf{U}^{dhp} , \mathbf{U}^{tfa}) or charged (\mathbf{U}^{am}) modifications showed moderate to strong inhibiting effects. Several previous studies^{52,66, 67, 68, 69, 70} and X-ray structure⁷¹ of T7 RNAP showed that specific major-groove interactions are more important in the template strand which is in accord with our findings that some bulkier or charged modifications may disrupt these interactions and inhibit transcription. Conversely, the non-template (sense) strand apparently offers more freedom for accommodation of modifications which are generally better tolerated and some of them may even positively contribute to the interactions with the polymerase and thus enhance the transcription. However, a deeper mechanistic and structural biology study would be needed to fully understand the effects.

We added references 51, 52, 66-71

Regarding the incorporation of synthetic nucleotide triphosphates by DNA polymerase, a discussion about the biosynthesis pathways of thymidine hypermodifications should be added to clarify the obtained results. Some DNA modifications naturally occur through different mechanisms, either before or after DNA polymerization. In this study, all modifications are introduced as triphosphate building blocks. It would be interesting to specify at which stage the pyrimidine modification takes place in cell among those studied here. If some of these modifications naturally occur post-polymerization, could that explain why they are less efficient substrates for DNA polymerase? Furthermore, do we know if such modified duplexes have already been synthesized from duplexes containing 5hmdU using specific enzymes involved in such hypermodification? Emphasizing this synthetic aspect would also further enhance the impact and significance of the study.

Response: we thank the reviewer for this valuable suggestion. Indeed, there is a correlation that needs to be discussed.

Revision: we added the following part to Discussion:

The comparison of our results with biosynthesis of viral DNA containing modified nucleobases shows a distinct correlation. The modifications that were easily incorporated even in PCR using modified **dU^xTPs** (**U^{hm}**, **U^{dhp}**) are biosynthetically available directly as nucleoside triphosphates, whereas **U^{glu}** and **U^{put}**, which failed to give full-length amplicon in PCR from their corresponding **dU^xTPs**, are introduced into DNA via post-synthetic transformations of **U^{hm}**.²⁰

Minor: a reference is lacking in Figure 1 (dT_{N3} synthesis)

Response: We thank the reviewer for noticing this mistake in the manuscript.

Revision: The reference (ref. 54) is now added.

Reviewer #2 (Remarks to the Author):

In this manuscript, the authors synthesized a series of dUXTPs derived from unnatural 5-substituted nucleosides, and performed polymerase assays (PEX, PCR) for these synthetic substrates using Vent and KOD DNA polymerases. Furthermore, they performed transcription assays using T7 RNAP with polynucleotides containing 5-substituted nucleosides as templates. Although the synthesis of these substrates is generally not easy, the substrates they synthesized this time have been satisfactorily identified by NMR and MS.

Accordingly, the reviewer recommends that the manuscript would be published with the following revisions:

-Assays of 5-substituted pyrimidine nucleoside triphosphates using thermostable DNA polymerases have already been reported (NAR, 2006, 5383), so comparisons and discussion are warranted in this paper (5-cyanomethyluridine/cytidine derivatives are also described in that report). Furthermore, the paper should also be cited.

Response: we thank the reviewer for this suggestion, and we agree.

Revision: An additional sentence has been added:

It should be noted, that incorporation of **dU^{cm}TP** in PCR using a similar mixture of thermostable DNA polymerases was reported previously.⁶⁰

Reference 60 was added

-Regarding 5-cyanomethylpyrimidine derivatives in microorganisms, a previous report (RNA. 2014, 177) should be cited and discussed.

Response: we thank the reviewer for this suggestion, and we agree.

Revision: An additional sentence has been added:

Interestingly, 5-cyanomethyluridine was found as a modification in tRNA of *Haloferax Volcanii*.³⁵

Reference 35 was added

-The sequences of poly-ribonucleotides generated by T7 RNAP should be statistically analyzed and the fidelity (accuracy) should be discussed (e.g., see JACS, 2020, 21530).

Response: We thank the reviewer for this suggestion. We agree that the assessment of fidelity could be relevant for the fully modified DNA templates (while we assume that for the promoter-only modified templates there is no reason why a non-modified gene would influence the polymerase fidelity). However, the transcripts in our study are too short (70 nts) for meaningful sequencing by RT and Sanger. Therefore, we NGS to due to the high accuracy and its ability to reliably show low percentage of mutations. We designed and performed a reverse transcription reaction, followed by preparation of NGS-ready DNA templates. We performed both adapter and index PCRs and the resulting NGS-ready templates were sequenced by Novogene. The results show no mutations in RNA samples transcribed from almost all DNA templates. Only in case of RNA coming from acetyl-dU-modified DNA template (**107DNA_F_U^{ac}**), several rA are partially replaced by rG in about 4% of all sequences. This result shows that acetylU is mutagenic (similarly to known formylU) and can pair with a guanosine.

Revisions: We added the following paragraph to Results:

Finally, we wanted to assess the fidelity of IVT with fully modified templates **107DNA_F** through sequencing of the produced RNA (**70RNA_F**). Therefore, we prepared samples for the next generation sequencing (NGS) by reverse transcription of purified **70RNA_F** from IVT, using M-MLV reverse transcriptase with target-specific primer **Prim_RT^{180N}** and template switching oligo **TSO^{300N}**, followed by PCR-based amplicon library preparations (see section 2.9. in SI for full description). Indexed libraries were pooled and sequenced on Illumina NovaSeq with an output of 2 millions of 2 x 150 bp paired-end reads per sample. The results (see Figures S145 – S150 in SI) show high transcription fidelity with templates **107DNA_F** containing natural U, or modified **U^{hm}**, **U^{et}**, **U^{She}** or **U^{cm}** nucleobases. Only in case of template containing the **U^{ac}** modification, the resulting RNA contained several partial (4-5%) A→G mutations (mainly positions 13, 16, 25, 43). Apparently, these mutations are caused by misincorporations due to mutagenic minor base-pairing between **U^{ac}** and G, similarly as has been previously reported for 5-formyluracil.⁶⁵ However, we cannot rule out the possibility that part of these mutations could have been introduced into the DNA during PCR with modified **dU^{ac}TP**.

We also added reference 65 on mutagenicity of 5-formylU:

65. Fujikawa, K., Kamiya, H. and Kasai, H. (1998) The mutations induced by oxidatively damaged nucleotides, 5-formyl-dUTP and 5-hydroxy-dCTP, in *Escherichia coli*. *Nucleic Acids Res.*, **26**, 4582–4587.

-Polymerases used in the experiments should be described in the footnotes of the figures.

Response: We agree.

Revision: We now refer to the used polymerases in the footnotes of figures where appropriate.

Reviewer #3 (Remarks to the Author):

The manuscript by Gracias et al. describes a continuation of previous work by the group on enzymatic synthesis of DNA templates containing novel and biologically relevant modified nucleotides. Subsequently the authors evaluated the influence of such modifications on the IVT with T7 RNAP. The work is highly important to the scientific community, with an emphasis on nucleic acid (bio)chemists and chemical biologists. The experiments are carried out well. I recommend for publication pending the revisions outlined below

1. Small molecule characterization should include 5 characterization techniques. The authors included ¹H, ¹³C and HRMS which are critical, but two additional techniques such as IR, UV, elemental analysis should be included for all novel compounds

Response: We agree that IR characterization should be included for all new nucleosides since it can confirm the presence of the additional functional groups. On the other hand, it is not necessary and feasible to perform this characterization on nucleoside triphosphates because they were all prepared by triphosphorylation of already fully characterized nucleosides and in small quantities that were needed for the biochemical experiments. Moreover, all modified dNTPs are also characterized by ³¹P NMR. UV characterization would not bring any useful information since none of the attached modifications has any significant effect on UV absorption, compared to their natural counterparts.

Revision: We added IR spectra for final nucleosides.

2. Scheme 1: missing Pd/C for i

Response: We thank the reviewer for noticing that this reagent is missing in the description.

Revision: We added the reagent to the footnotes of Figure 3.

3. Page 5 line 102, I could not find dUsm, is this dUasm? If not, a chemical structure should be drawn in one of the figures or schemes for reference (note that Scheme 4 might be too far in the text for proper reference)

Response: dU^{sm} and dU^{asm} are indeed two different compounds. We have not synthesized compound dU^{sm} (only its protected form), but we agree that the structure should be shown for reference.

Revision: A structure of dU^{sm} is now shown in Figure 3, stating that it was not prepared as free nucleoside but rather as Ac-protected dU^{asm} .

4. I am interested in the concomitant deprotection of Boc and Ac in MeOH. It would seem to me that upon addition of AcCl, the amine beta to the nucleobase would undergo acylation. The amine is more nucleophilic than methanol, and I would expect major by product formation with this step. Can the authors comment on this?

Response: There is certainly some acylated by-product formed, but we did not try to isolate or quantify it. However, the big excess of MeOH as the solvent reduces this unwanted reaction, as well as immediate protonation of secondary amine by *in situ* generated HCl.

Revisions: We changed the following sentence to:

Concomitant deprotection of both acetyl- and Boc- groups by acetyl chloride in dry methanol provided dU^{put} (20% in 4 steps from thymidine) as the major product, which again needed to be TFA-protected before triphosphorylation.

5. Page 7: authors mention low yield for certain triphosphates. Is there a reason for this? Can the authors offer a bit more information as to why this could be the case for these compounds.

Response: The reason for the low yield of $dU^{glu}TP$ is sufficiently described. Yields of $dU^{tfa}TP$ (and $dU^{am}TP$ prepared from the same nucleoside as $dU^{tfa}TP$) were low in all three performed triphosphorylations (one for $dU^{tfa}TP$, two for $dU^{am}TP$). We believe that a partial deprotection of TFA group takes place, followed by phosphorylation of the deprotected primary amine.

Revision: We added the following text into the manuscript:

The yields ranged from 3 to 30%. The lowest yields (3% for $dU^{tfa}TP$ and 7% for $dU^{am}TP$) were achieved in triphosphorylation of dU^{tfa} , which showed numerous side-products, probably due to partial deprotection and subsequent reactivity of the primary amine.

6. An additional Figure should be generated for the main text concerning the initial primer extension experiments (including the generation of the 107 bp template). It is difficult to follow the discussion of about 1.5 pages, when always referring to the supporting information.

Response: We agree with the reviewer that an additional figure depicting initial testing of enzymatic incorporations should be added.

Revision: We added a figure (Figure 7), summarizing initial testing of enzymatic incorporations.

7. Can the authors provide more explanation concerning the Utfa providing enhanced sense-strand activity and strong inhibition on antisense-strands?

Response: Also based on the suggestion of Reviewer 1, we added the discussion of the different effect of modifications in template and non-template strands.

Revision: We added the following part to the Conclusion:

On the other hand, the modifications in the coding (antisense) strand showed higher differences where some nucleobases bearing small neutral modifications had slightly stimulating (U^{sm} , C^{hm} , U^{hm}) or neutral (U^{et} , C^{et} , U^{ac} , C^{ac}) effect, whereas bulkier (U^{dhp} , U^{tfa}) or charged (U^{am}) modifications showed moderate to strong inhibiting effects. Several previous studies^{52,67,68,69,70,71} and X-ray structure⁷² of T7 RNAP showed that specific major-groove interactions are more important in the template strand which is in accord with our findings that some bulkier or charged modifications may disrupt these interactions and inhibit transcription. Conversely, the non-template (sense) strand apparently offers more freedom for accommodation of modifications which are generally better tolerated and some of them may even positively contribute to the interactions with the polymerase and thus enhance the transcription. However, a deeper mechanistic and structural biology study would be needed to fully understand the effects.

We also added new references 67-72

8. There seems to be inconsistency in the abbreviations used for the chemical modifications. E.g. what is the difference between Ucm and dUcm. One would logically think that the one with the “d” is DNA and the U alone is RNA. Is this the case? Since there are so many abbreviations in the text, the authors should really try to clarify this terminology.

Response: Abbreviations dU^x describe the whole nucleoside. Abbreviations U^x then refer to the modified nucleobase. However, we agree that further explanation of these abbreviations would make the text clearer.

Revisions: In Figure 7 and Figure 8, we added the following:

N = natural DNA, **U^x** refer to modified nucleobases.

Response to reviewers - COMMSCHEM-24-0389A

REVIEWERS' COMMENTS:

Reviewer #1 (Remarks to the Author):

The authors have made appropriate revisions to the manuscript and are congratulated on the advances reported.

Reviewer #2 (Remarks to the Author):

The manuscript has properly been revised.

Reviewer #3 (Remarks to the Author):

The authors have addressed all my comments (and other reviewer comments, in my opinion).

Response: we thank the reviewers for the positive evaluation of our revised manuscript